# Clinical Trials of New Drugs for Vascular Cognitive Impairment and Vascular Dementia

**DOI:** 10.3390/ijms231911067

**Published:** 2022-09-21

**Authors:** Tran Thanh Duy Linh, Yi-Chen Hsieh, Li-Kai Huang, Chaur-Jong Hu

**Affiliations:** 1International Ph.D. Program of Medicine, College of Medicine, Taipei Medical University, Taipei 110, Taiwan; 2Family Medicine Training Center, University of Medicine and Pharmacy at Ho Chi Minh City, Ho Chi Minh City 700000, Vietnam; 3Ph.D. Program in Medical Neuroscience, College of Medical Science and Technology, Taipei Medical University, Taipei 110, Taiwan; 4Master’s Program in Applied Epidemiology, College of Public Health, Taipei Medical University, Taipei 110, Taiwan; 5Ph.D. Program in Drug Discovery and Development Industry, College of Pharmacy, Taipei Medical University, Taipei 110, Taiwan; 6Dementia Center and Department of Neurology, Shuang Ho Hospital, Taipei Medical University, New Taipei 235, Taiwan; 7Taiwan Taipei Neuroscience Institute, Taipei Medical University, Taipei 110, Taiwan; 8Graduate Institute of Medical Neuroscience, College of Medical Science and Technology, Taipei Medical University, Taipei 110, Taiwan; 9Department of Neurology, School of Medicine, College of Medicine, Taipei Medical University, Taipei 110, Taiwan

**Keywords:** vascular dementia, vascular cognitive impairment, clinical trial

## Abstract

Population aging has challenged the treatment of cognitive impairment or dementia. Vascular dementia is the second leading cause of dementia after Alzheimer’s disease. Cognitive consequences after ischemic brain injury have been recognized as a preferred target for therapeutic strategies, prompting the search for potential agents. The keyword “vascular dementia” was used to search ClinicalTrials.gov to determine agents represented in phases I, II, III, and IV. The agents were classified on the basis of their mechanisms. Of the 17 randomized controlled trials meeting our inclusion criteria, 9 were completed in the past 10 years, and 8 are ongoing or in the planning stages. We also identified one trial in phase I, nine in phase II, six in phase III, and one in phase IV. Fewer trials of new drugs for improving cognition or ameliorating the behavioral and psychological symptoms of dementia target vascular dementia than Alzheimer’s dementia. Drug trials on vascular dementia overlap with drug trials targeting functional outcomes in cerebrovascular disease. International pharmaceutical companies’ investment in new drugs targeting VCI and vascular dementia remains insufficient.

## 1. Introduction

### 1.1. Definitions of Vascular Cognitive Impairment and Vascular Dementia 

Vascular cognitive impairment (VCI) refers to cognitive impairment beyond the normal aging process and the underlying age-related vascular disease [1,2]. VCI covers all states of cognitive impairment associated with the vascular disorder, from mild cognitive deficits to dementia [1]. Subtypes of VCI are classified on the basis of the degree of cognitive decline, which ranges from the impairment of a single cognitive domain to overt vascular dementia (VaD) which indicates activities of daily life affected by cognitive decline. VaD has been used regardless of whether the pathogenesis of vascular lesions is ischemic or hemorrhagic (characteristic of poststroke cognitive impairment; PSCI) or single or multiple infarcts. Furthermore, as VaD combines different vascular mechanisms and changes in the brain, and has different etiologies and clinical presentations, this heterogeneity in definition can affect the outcomes of clinical trials. Therefore, subcortical (ischemic) VaD is a more homogeneous population and may be an alternative for clinical drug trials, including small vessel disease, lacunar cerebral infarction, and ischemic white matter lesions [3].

### 1.2. Epidemiology of VCI

A meta-analysis of studies with various methodologies and diagnostic criteria indicated that the pooled prevalence for all-cause dementia, AD, and VaD was 697, 324, and 116 per 10,000 persons, respectively [4]. VaD accounts for at least 20–40% of all dementia diagnoses. Because of differences in the definition of PSCI and study settings, the prevalence of PSCI in the literature varies from 22% to 58% [5,6]. Dementia prevalence in the first year after stroke ranged from 7% in population-based studies of first strokes (excluding prestroke dementia) to 41% in hospital-based studies (including recurrent stroke and prestroke dementia) [7]. In hospital-based and community-based studies, after the inceptive incidence of poststroke dementia, the cumulative incidence increased linearly by 3% and 1.7% per year, respectively [7]. A population-based study with 25 years of follow-up observed a cumulative incidence of poststroke dementia of 48% [8]. Risk factors for VCI overlap considerably with those for stroke, the most salient of which is increasing age [9,10]. Other non-modifiable risk factors include female sex and heredity [9,11]. Protective modifiable risk factors comprise higher education, occupation, social networks, cognitive and physical activity, and following a Mediterranean diet [1,12,13]. Vascular risk factors such as hypertension, diabetes, and obesity increase the risk of VCI [1,12]. Depression later in life is also associated with a higher risk of all-cause dementia, AD, and VaD [14].

### 1.3. Unmet Needs

As population aging progresses on a global scale, dementia has emerged as a critical public health concern. Cognitive decline occurs before dementia symptoms manifest. Mild cognitive impairment associated with vascular lesions is the preferred target for therapeutic strategies aimed at slowing or halting cognitive decline such that progression to dementia and the related loss of autonomy can be avoided. The maintenance of vascular health and the preservation of brain function can mitigate the negative effects of dementia on public health. It can also provide insight into the pathobiology, prevention, and diagnosis of this disease [15,16]. However, no specific treatment has been successfully developed, prompting the introduction of the disease modifier concept.

## 2. Pathophysiology of VCI and VaD

The pathophysiology of brain injury in VCI or VaD is complex, involving multiple neuronal and vascular pathologies. Proposed mechanisms include atherosclerosis, cerebral small-vessel diseases, cerebral hypoperfusion, oxidative stress and inflammation, endothelial dysfunction, and blood–brain barrier (BBB) disruption [17]. Emerging evidence has suggested that cholinergic degeneration contributes to clinical cognitive decline in VCI or VaD [18,19,20] (Figure 1). 

### 2.1. Atherosclerosis and Cerebral Small-Vessel Diseases

Atherosclerosis is a chronic inflammatory condition characterized by the accumulation of lipoprotein and fibrous elements beneath injured endothelial cells as well as by the involvement of macrophages and lymphocytes in plaque formation. Atherosclerosis often affects large and medium arterial vessels [21] such as the aorta, carotid, and intracerebral arteries. The substantial accumulation of plaque can block blood flow, leading to a stroke [22]. Cerebral infarcts or reduced cardiac output caused by myocardial infarction can induce cerebral hypoperfusion [23,24] and forward-altering cognitive function subsequently [25]. The types of VaD related to atherosclerosis are typically large-vessel or multi-infarct dementia [26].

The most common vascular contributor to dementia is cerebral small-vessel disease (SVD), which refers to pathological changes in the penetrating and perforating arterioles of the brain. Pathological changes induced by SVD range from hyaline deposition and hyperplastic arteriolosclerosis to vessel fibrosis, which causes microvascular stiffening and distortion, damaging the BBB and leading to lacunar infarcts, microinfarcts, and white matter demyelination [26,27].

### 2.2. Hypoperfusion

The brain depends on a continuous blood supply to provide the large amount of energy required to maintain its structural and functional integrity [28]. Therefore, cerebral hypoperfusion has a critical pathophysiological contribution to VCI or VaD. Hypoperfusion leads to cumulative brain tissue damage resulting from hypoperfusion-associated injuries such as white matter injury, lacunar infarcts, brain atrophy, microbleeds, and microinfarcts [29]. Chronic cerebral perfusion (CBP) reduction can be caused by carotid artery stenosis, cerebral microvasculature blocks, or global hemodynamic dysfunction due to heart failure, cardiac arrest, or hypotension [30]. These conditions have been revealed to induce brain dysfunction and cognitive impairment transiently or permanently [31,32,33]. 

### 2.3. Oxidative Stress and Inflammation

Markers of oxidative stress and inflammation, specifically microglial and astrocyte activation and elevated cytokine levels, have been observed in white matter lesions associated with VCI [34,35,36]. These responses may be triggered by hypoxic–ischemic encephalopathy resulting from chronic hypoperfusion. Peroxidation products and free radicals generated during oxidative stress and gliosis in inflammation have been postulated to alter vascular permeability and induce neurotoxicity, resulting in the loss of white matter integrity [37,38]. Increased oxidation and inflammation may increase susceptibility to atherosclerosis [39,40], accelerating neurodegeneration.

### 2.4. Endothelial Dysfunction and Altered Blood–Brain Barrier (BBB) Permeability

Cerebral endothelial cells are susceptible to the attack of hypoperfusion. Under oxidative stress and inflammation, the endothelial nitric oxide synthase (eNOS) pathway is impaired, reducing nitric oxide (NO) production and bioreactivity (eNOS/NO signaling), and leading to endothelial dysfunction. These dysfunctional endothelial cells alter the permeability of the BBB. Consequently, neurons are exposed to toxic substances, and neurovascular trophic coupling is disrupted, exacerbating cerebral hypoperfusion in response to brain activity. Endothelial dysfunction and cerebral hypoperfusion affect each other. Considering the coexistence of VaD risk factors such as aging, obesity, and hypertension, the pathogenesis of VCI or VaD is characterized by a vicious circle [41].

### 2.5. Cholinergic Hypothesis

The premise of cholinergic involvement in VaD was suggested on the basis of findings from several studies. Markedly disturbances of the cholineacetyl transferase were found early in a 1989 study of brain tissues among dementia patients with histories of stroke [18]. In another study, the degeneration of cholinergic nuclei in the prosencephalon and the derangement of their projections were observed in patients with mixed dementia [19]. Another study of post-mortem brain tissue observed a notable loss of several types of cholinergic neurons in the cortex and hippocampus of patients with VaD [20]. This reduction was later associated with cognitive impairment and correlated with white matter hyperintensity in magnetic resonance imaging (MRI) [19,20], but the mechanism for this deficiency has remained unclear. A cholinergic deficit is caused when focal, multifocal, or diffuse vascular and ischemic lesions are involved in brain structures or regions (e.g., the basal ganglia, thalamus, white matter, and subfrontal area) and interrupt frontostriatal circuits. Cholinergic dysfunction resembles that observed in patients with AD due to a dense network of cholinergic fibers in the injured area [42,43,44].

## 3. Results

The keyword “vascular dementia” was used to search ClinicalTrials.gov to determine agents represented in phases I, II, III, and IV. The agents were classified on the basis of their pathophysiological mechanisms. Of 17 randomized controlled trials (RCTs) meeting our inclusion criteria, 9 were completed in the past 10 years, whereas 8 are ongoing or in the planning stages. We identified one trial in phase I, nine trials in phase II, six trials in phase III, and one trial in phase VI (Table 1, Table 2 and Table 3 and Figure 2).

### 3.1. Antioxidant and Anti-Inflammatory Agents 

DL-3-n-butylphthalide (NBP) is a synthetic drug developed from L-3-n-butylphthalide, a natural compound extracted from celery oil. NBP exerts antioxidant, antiapoptotic, and antithrombotic effects. It also protects against mitochondrial damage. The drug was approved in China for treating ischemic stroke–induced neuronal impairment [45]. It is also used to treat Alzheimer’s disease (AD) and Parkinson’s disease [46]. NBP improves VCI or VaD through its protective effects of oxidative stress suppression, neuronal apoptosis inhibition, and the reduction of beta-amyloid (Aβ) deposits [47]. In 2016, an RCT revealed that 6-month treatment with NBP improved cognitive function in patients who had subcortical ischemic SVD without dementia [48]. A systematic review of 26 studies indicated that NBP was effective in enhancing cognitive function and the ability to perform activities of daily living (ADLs) after stroke [49]. In 2019, a phase III trial (NCT03804229) investigated the 52-week use of butylphthalide soft capsules (equivalent to 600 mg of NBP per day divided into three portions) on patients with VaD. The phase III trial had established the effectiveness of the drug. This RCT is recruiting patients aged between 50 and 75 years who meet the criteria for having a major vascular neurocognitive disorder as listed in the *Diagnostic and Statistical Manual of Mental Disorders, Fifth Edition* (DSM-5). Patients with dementia caused by other cerebral conditions (e.g., AD or brain tumors) or severe comorbidities (heart or lung diseases) will be excluded. The effectiveness of the agents will be assessed on the basis of improvement in scores on psychoneurological assessments, namely the vascular dementia assessment scale cognitive subscale (VaDAS-cog), clinician interview–based impression of severity (CIBIC-plus), alzheimer’s disease cooperative study—activities of daily living, neuropsychiatric inventory (NPI), and mini-mental state examination (MMSE).

SaiLouTong (SLT) is a traditional Chinese medicine consisting of *Ginkgo biloba*, *Panax ginseng*, and *Crocus sativus* (saffron) extracts in a specific dose ratio. This combination was determined on the basis of the pharmacological (antioxidant, anti-inflammatory, and blood flow enhancing) effects of these three herbs on VaD such that cerebral hypoperfusion can be ameliorated. Ginsenosides have been demonstrated to reduce amyloidβ (Aβ) and cholinesterase activity [50]. A series of phase I and phase II trials on SLT has demonstrated its safety and effectiveness in improving cognitive and memory function as well as auditory and speech processing [51,52,53,54]. A phase III trial involving the treatment of mild-to-moderate VaD with SLT (NCT03789760) that began in 2019 and aims to validate the promising effects of SLT. SLT is expected to improve cognitive and executive functions, the ability of ADLs, and psychological behaviors. Recruiting is ongoing, and 2023 should be completed.

Tianzhi granules (TZ) are an herbal medicine approved by the China Food and Drug Administration (CFDA) for VaD treatment. The main components of TZ are *gastrodin*, *geniposide*, *rutin*, and *baicalinand*. TZ was demonstrated to mitigate oxidative stress, apoptosis, and necrosis induced by chronic cerebral hypoperfusion in a rat model [55,56]. An RCT conducted in 2020 (NCT02453932) indicated that TZ and donepezil, an anti-AD agent, exert the same therapeutic effects on cognitive function and BPSD in patients with mild-to-moderate VaD. Specifically, the CIBIC-plus and NPI scores in the TZ group were significantly higher than those in the placebo group (*p* < 0.001) and did not differ from those in the donepezil group. However, the study was limited by the fact that the placebo group was smaller than the TZ and donepezil groups as well as with strong placebo effects [55]. 

N-acetylcysteine (NAC) is a precursor of L-cysteine, a glutathione (GSH) component critical to endogenous antioxidant activities and immunity. NAC confers the potential to promote cognitive function and slow the progression of dementia through its antioxidant characteristic. This agent scavenges free radicals and alleviates oxidative stress by maintaining or increasing GSH levels. Animal studies have demonstrated the neuroprotective and cognitive-enhancing effects of NAC [57,58]. In humans, taking 600 mg of NAC daily for 6 months improves the dementia rating scale and protects against executive function impairment in mild cognitive impairment [59]. The same finding was revealed in a double-blind RCT conducted on patients with AD with the same dose of NAC [60]. To determine the effectiveness of NAC on vascular-related cognitive impairment, a Canadian research group conducted a phase II trial (NCT 03306979) in which patients with VCI were randomly assigned to take NAC or a placebo for 24 weeks. The dose of NAC was maintained as high as 2400 mg per day in the first and third weeks and at 1200 mg in the second and last weeks. The NAC supplement was considered an add-on therapy to improve cognitive function in patients enrolled in a cardiac rehabilitation program. This study was completed in 2018, but its data have not yet been published.

BAC is a new agent developed by CharSire Biotechnology Corporation in Taiwan to treat VaD and other diseases. BAC is a vapor fraction from seeds of *Glycine max (L.) Merr,* also known as soybeans. BAC has been demonstrated to mitigate cognitive impairment in murine models of ischemic stroke through its action on inflammation in the brain. Specifically, BAC significantly reduced levels of procytokine interleukin 1-β, which is believed to damage brain tissue. Preliminary data from a phase II RCT (registered to NCT02886494) indicated that BAC benefits cognitive function, neuropsychiatric behaviors, and ADL scores among specific dementia populations, including individuals with mixed type dementia or individuals naïve to dementia medication [61]. 

### 3.2. Agents to Mitigate Endothelial Dysfunction 

CY6463 has been developed as a promising therapeutic agent for several neurodegeneration diseases. As a guanylyl cyclase (sGC) stimulator, CY6463 penetrates the BBB. It amplifies the activity of the nitric oxide–sGC–cyclic guanylate monophosphate pathway, which is impaired in cognitive impairment and dementia [62]. The pathway is known to modulate brain blood flow, neuroinflammation, and vascular tone and has been implicated in neuronal function [63,64]. The pharmacological effect of CY6463 may stem from the compensation of NO deficiency, thereby restoring endothelial function, enhancing cerebral perfusion, and improving cognitive function [65]. In 2021, a phase II RCT (NCT04798989) was being conducted to determine the safety of CY6463 in patients diagnosed with AD and vascular pathology. The recruitment process is ongoing.

### 3.3. Multitarget Agents

Tianmabianchunzhigan (TMBCZG), a CFDA-approved compound extracted from *Gastrodia elata* which is used to treat VaD in traditional Chinese medicine. TMBCZG affects VaD because gastrodin, one of its components, inhibits inflammation, autophagy, and apoptosis in rat models and suppresses Aβ formation [66]. In 2017, Tian et al. conducted a multicentered phase IIa trial (NCT03230071) on 160 patients with VaD to compare the efficacy and safety of TMBCZG over 24 weeks of treatment. The patients were randomly assigned to receive a high-dose regimen (84 mg per day), low-dose regimen (28 mg per day), or placebo. The study was completed in 2021 and followed by an active phase IIb trial (NCT05371639) in 2022 by the same research group, which is in the recruitment stage. In the second trial, the duration of VaD treatment will be extended to 36 weeks, with a high dose of TMBCZG or placebo. Both trials recruited patients who were aged between 55 and 80 years and met the diagnostic criteria for VaD of the National Institute of Neurological Disorders and Stroke and the Association internationale pour la recherche et l’enseignement en Neurosciences (NINCDS-AIREN). The effectiveness of the agents is determined on the basis of improvement in VaDAS-cog, CDR sum of box (CDR-SB), MMSE, and ADL scores. 

Fufangdanshen tablets (FFDS) is a traditional Chinese medicine approved by the CFDA to treat VaD. The main components of FFDS, are extracted from *Salvia miltiorrhiza*, *Panax notoginseng*, and *Borneolum syntheticum*, include tanshinone, salvianolic acid, ginsenosides, and borneol. FFDS exerts multiple effects on dementia through the specific pharmacological effects of its components. For example, tanshinone inhibits iNOS and MMP 2, reduces free radicals, and eliminates oxidative stress [67]. Salvianolic acid targets oxidation markers and suppresses glial activation and the production of inflammatory cytokines [68], whereas borneol increases BBB permeability [69]. Preclinical studies on rodents have indicated that FFDS can enhance cognitive and memory function [70]. In humans, a phase II clinical trial conducted on patients with VaD revealed the favorable efficacy of FFDS in improving MMSE and ADLs. However, the study was limited by bias attributed to the lack of placebo comparison, a small sample size, and a short follow-up duration (12 weeks). In 2012, another phase II randomized trial (NCT01761227) was designed with a control group. The outcomes of 254 patients meeting diagnostic criteria for VaD in accordance with the DSM-5 were examined. The primary outcomes were changes in the scores on the alzheimer’s disease assessment scale cognitive subscale (ADAS-cog) as well as changes in scores on the CIBIC-plus, MMSE, and ADL scales. Additional RCTs are warranted to determine the safety and efficacy of FFDS in the treatment of VCI and VaD. 

Metformin, a standard antidiabetic drug, may be promising for treating VCI and VaD and other types of dementia. Its effects are due to molecular action that ameliorates oxidative stress and inflammation, in addition to its hypoglycemic properties. In both in vitro and in vivo studies, metformin has been demonstrated to scavenge hydroxyl free radicals and reduce the expression of activated glial markers, inflammation markers, and interleukins [71]. Metformin also improves endothelial function in the adenosine monophosphate–activated protein kinase–dependent pathway, preventing vascular events complicated by diabetes [72]. The inhibition of AChE by metformin can be an additional effect on VaD despite inconsistent findings in studies [73,74]. However, only two RCTs on metformin have revealed its advantages over a placebo in terms of cognitive function and cerebral blood flow [75,76]. In 2013, a phase II clinical trial (NCT01965756) was conducted to investigate the effect of metformin on MCI and dementia caused by vascular pathology or AD. Twenty patients were assigned to receive metformin or a placebo for 8 weeks, and they then crossed over to the other intervention for the next 8 weeks. The trial was completed in 2017, and its preliminary data suggest that metformin tends to improve memory and neurophysiological outcomes.

### 3.4. Agents for Restoring the Central Cholinergic or Glutamatergic System

Donepezil, a cholinesterase inhibitor, slightly improves cognitive function in patients with VCI. In a systematic review and Bayesian network meta-analysis, 10 mg of donezepil caused stable moderate improvement and was statistically superior to the placebo in terms of both MMSE and ADAS-cog scores [77]. Wilkinson et al. traced participants with possible or probable VaD in an international, multicenter, open-label, 30-week extension study. Donepezil improved cognition (on the basis of ADAS-cog scores) for up to 54 weeks in patients with VaD [78]. The 2011 American Heart Association and American Stroke Association guidelines recommend that patients with VaD undergo donepezil treatment for cognitive benefits. However, evidence for global and functional efficacy from donepezil is less consistent [1]. A 24-week, multicenter, double-blind RCT discovered that rivastigmine did not deliver consistent efficacy in improving the ability to perform ADLs or in mitigating neuropsychiatric symptoms in patients with probable VaD. The effectiveness of rivastigmine on cognitive outcomes was determined through an examination of its effects on older patients likely to have concomitant Alzheimer’s pathology [79]. A Cochrane Library review concluded from the data of three trials that rivastigmine has some benefits for VCI. Because of differences in study designs, no pooling of study results was attempted [80]. A multinational, double-blind, randomized, placebo-controlled trial reported that galantamine was effective in improving cognition in patients with VaD, as assessed using CIBIC-plus scores. However, ADL scores after galantamine treatment were similar to those after a placebo [81]. In 2020, the fifth Canadian Consensus Conference has recommended cholinesterase inhibitors may be used to treat vascular cognitive impairment in selected patients who are explained about the benefits and harms of these drugs [82]. In this review, we found a phase I, randomized, single group, open-label study (NCT00457769) which aims to determine whether donepezil improves the recollection of the steps of functional tasks. This study has a small sample size of only 18 participants. The study was first registered on ClinicalTrials.gov in 2007. On 16 February 2021, its status was updated to active but not recruiting. Although the study title indicates that vascular dementia is its focus, no clear definition of the diagnostic criteria for case enrollment is presented.

A Cochrane Library review of two studies of approximately 750 participants concluded that memantine confers small clinical benefits for cognitive function with low-to-moderate certainty. The numbers of individuals experiencing adverse events in the memantine and placebo groups were similar, and the numbers of individuals discontinuing treatment were also similar between the groups. A post hoc subgroup analysis of severity suggested that memantine had a more considerable effect on cognitive function in people with moderate-to-severe VaD (MMSE score of ≥14) than on cognitive function in people with mild-to-moderate VaD [83]. One phase III clinical trial conducted in Russia (NCT03986424) aims to evaluate the clinical efficacy and safety of 20 mg of akatinol memantine (single doses) versus 10 mg of akatinol memantine (double doses) in patients with moderate and moderately severe vascular dementia, MMSE scores of 10–20, and a Hachinski ischemic score of ≥7 points. It is expected to enroll 126 participants, and the estimated date of completion is December 2022. The primary end point is a change in total ADAS-cog score from baseline to after 24 weeks of use. 

### 3.5. Agents for Treating Behavioral Psychological Symptoms of Dementia (BPSD)

BPSD, as indicated by the International Psychogeriatrics Association, is also referred to as neuropsychiatric symptoms of dementia, including changes in behavior, perception, thoughts, and disordered mood [84]. Fuh et al. studied neuropsychiatric profiles in patients with AD and VaD in Taiwan. A total of 536 participants (161 with subcortical VaD, 35 with cortical VaD, and 16 with mixed cortical and subcortical VaD) were recruited. Patients with cortical VaD had the highest mean composite NPI scores in all domains. Patients with cortical VaD and subcortical VaD scored higher in apathy than did patients with AD [85]. A Swedish registry study on cognitive disorders and dementia revealed that individuals with VaD had a higher risk of apathy, but a lower risk of agitation or aggression, anxiety, and aberrant motor behavior. Agitation and aggression are more relevant to mixed-type dementia than to VaD [86]. A study conducted at three major medical centers in Taiwan enrolled 97 patients with BPSD. Probable VaD lasting ≥3 months poststroke was diagnosed on the basis of the criteria of the NINCDS-AIREN. In this randomized, double-blind, placebo-controlled drug trial, participants were allocated randomly to interventions with NMDA enhancers, sodium benzoate or a placebo for 6 weeks. Sodium benzoate treatment improved cognitive function (in terms of ADAS-cog scores) only in women with later-phase dementia (15). Sodium benzoate exerted antipsychotic properties in patients with schizophrenia [86]. However, it did not mitigate psychotic symptoms in patients with dementia. The authors determined that the reason for this was that the dosage was considerably lower than that used in studies on schizophrenia [87]. Pimavanserin is a serotonin receptor modulator that acts primarily as a selective 5-hydroxytryptamine receptor subtype 2A inverse agonist and antagonist. A double-blind, placebo-controlled phase III discontinuation trial of pimavanserin prescribed to treat hallucinations and delusions associated with dementia-related psychosis enrolled participants with all-cause dementia, of which 9.7% had vascular dementia. Among the 217 participants who underwent randomization after 12 weeks of the open phase, the percentage of patients who had a psychotic relapse was 13% among those who continued to receive pimavanserin and 28% among those who were switched to placebo (approximate difference: 16%) [87]. A randomized, placebo-controlled, double-blind, parallel-group, multicenter phase II trial (NCT01608217) evaluated the efficacy and safety of low-dose delta-9-tetrahydrocannabinol (THC) in behavioral disturbances and pain in patients with mild-to-severe dementia. The study enrolled 50 participants with possible or probable dementia, including VaD or mixed-type dementia, on the basis of National Institute of Neurological and Communicative Disorders and Stroke and the Alzheimer’s Disease and Related Disorders Association (NINCDS–ADRDA) or NINCDS-AIREN criteria or the opinions of an expert panel. The participants were required to have clinically relevant BPSD (minimum NPI score of ≥10), with reported symptoms of agitation, aggression, or aberrant motor behavior present for at least 1 month prior to the screening. No benefits on behavioral disturbances, ADLs, pain-related behavior, or pain intensity in patients with dementia were conferred by 4.5 mg of oral THC taken daily over 3 weeks. However, THC was safe and well tolerated [88].

## 4. Discussion

Fewer new drug trials target the amelioration of cognitive impairment or BPSD in VCI than in Alzheimer’s dementia (Figure 3). Trials on drugs for treating VCI overlap with trials on drugs targeting functional outcomes in cerebrovascular diseases with respect to disability rates, modified Rankin scale scores, Barthel index scores, or mortality rates. Therefore, cognitive outcomes do not outweigh the importance of the recovery of motor function and the ability to perform ADLs, which are seldom evaluated as major independent outcomes. The primary end points of several trials were scores on the EuroQol-5 dimension (EQ-5D), Montreal cognitive assessment (NCT03759938 and NCT05046106), and verbal learning test (NCT04854811), as well as language production ability as assessed using lexical features of discourse in the cookie theft picture description (NCT0343463). As in a review of the VCI mechanism, heterogenicity and multiple pathophysiologies may be the main challenge. In a pathological study of 4429 individuals with clinically diagnosed AD, 80% had vascular pathology [89]. The co-occurrence of cardiovascular disease lowers the threshold for dementia caused by a single neurodegenerative process. Narrowing down the pathogenetic mechanism of AD to disease-specific mechanisms is impossible. No surrogate fluid biomarker has been established for probing the underlying mechanisms. The benefits of drugs targeting specific functions may be limited in other aspects. Some trials progress from bedside to bench by applying natural compounds or compounds conventionally used in traditional Chinese medicine to relevant clinical scenarios. 

A phase III trial on butylphthalide soft capsules (NCT03804229) is in the recruitment phase. The active ingredient is a compound derived from the seeds of *Apium graveolens*. A phase II study (NCT02886494) on BAC, derived from *G. max (L.) Merr*, is an example of a botanical drug developed to treat VCI. A phase III trial (NCT03789760) in the recruitment phase (with an expected sample size of 500) aims to administer SaiLuoTong capsules (120 mg) twice a day, 0.5 h before breakfast and dinner, over 52 weeks. The primary outcome is scored on the VaDAS-cog and Alzheimer’s Disease Cooperative Study–Clinical Global Impression of Change after treatment ends. A phase III trial of TZ granules (NCT02453932) was completed in 2017. Of the 543 patients with mild-to-moderate VaD, 242 took TZ granules, 241 took donepezil, and 60 took a placebo. Improvement in the CIBIC-plus was 73.71% and 58.18% in the TZ and placebo groups, respectively. This between-group difference was significant (*p* = 0.004). These are two examples of compounds conventionally used in traditional Chinese medicine, which can also be taken as a cocktail therapy, simultaneously acting on multiple proposed mechanisms. 

In this review, we focused on trials registered to ClinicalTrials.gov and did not expand our search to other databases such as AMIce, a German drug information system ChicCTR; a Chinese clinical trial register; CTRI, an Indian clinical trials register; or ALOIS, a database of Cochrane Collaboration Dementia and Cognitive Impairment Group might be a limitation. A review by Smith et al. in 2017 also summarized drug development for vascular dementia but by a search on ALOIS [90]. They identified 130 RCTs from 1966 to 2016 that preceded our trials’ timeline by forty years. They reported more trials than we did since they included both pharmacologic and non-pharmacologic interventions, and not only for treatment purposes but also for preventative aspects. The authors also classified the trials based on the drugs’ therapeutic effects on VCI pathophysiology, however, they defined some different drug classes including vasodilators, neurotrophic, antithrombotic, lipid-lowering, and metabolic-based mechanisms. The differences can be explained by the different concepts about the diseases’ pathophysiology. Regarding the pipeline of VCI drug development, Smith’s review stated a predominance in trials targeting perfusion enhancement via vasodilators which are popular in the first twenty decades, while later, especially from 1990 to 2016, more trials focused on testing the drugs classified as neurotransmitter modulators or multiple mechanisms of action. This finding, somehow, is in line with our report on emerging trials on multi-targeted agents. 

In short, the trend of drug development in VCI and VaD treatment has changed over time based on a growing understanding of the disease’s pathophysiology and the advances in diagnosis and measurement. However, international pharmaceutical companies’ investment in new drugs targeting VCI is insufficient. 

## 5. Materials and Methods

This review summarized the drug development pipeline for VCI and dementia. We used the keyword “vascular dementia” to search ClinicalTrials.gov for relevant trials meeting our inclusion criteria (Figure 4).

We presented the phases and status of the trials and drug development for VCI and VaD treatment. We also described the agents in our selected trials in terms of their therapeutic mechanism of action on the pathophysiology of VCI and VaD. Agents targeting several pathophysiological mechanisms were classified as multitarget. If the mechanism could not be identified in the literature, the mechanism was labeled as unknown. 

### 5.1. Type of Trials

We included all relevant RCTs in phase I, II, III, and IV. On the basis of the status of these trials on ClinicalTrials.gov, we included trials that were active but not recruiting; recruiting; enrolling by invitation; and not yet recruiting. We also included trials completed between 2012 to 2022.

We excluded trials that were terminated, withdrawn, unknown, suspended, or completed before 2012. We also excluded trials with non-RCT study designs, such as case–control studies and cohort studies.

### 5.2. Type of Participants

We included participants of all ages and both sexes with diagnoses of VaD or VCI without dementia. We also included trials including those with dementia or vascular diseases as long as a VaD subgroup was included.

We excluded trials that only examined Alzheimer’s dementia, stroke, and heart failure.

### 5.3. Types of Interventions

We included all pharmacological trials. We excluded trials for nontreatment purposes, such as diagnosis, prevention, or screening. We did not include nonpharmacological trials, such as those examining behavioral therapies, procedures, or devices.

### 5.4. Type of Outcome

Cognitive improvement was the primary outcome.

Cognitive function was measured using standardized tools.

Improvements in white matter integrity were determined through imaging techniques, such as MRI.

The mitigation of neuropsychiatric symptoms was the secondary outcome.

## Figures and Tables

**Figure 1 ijms-23-11067-f001:**
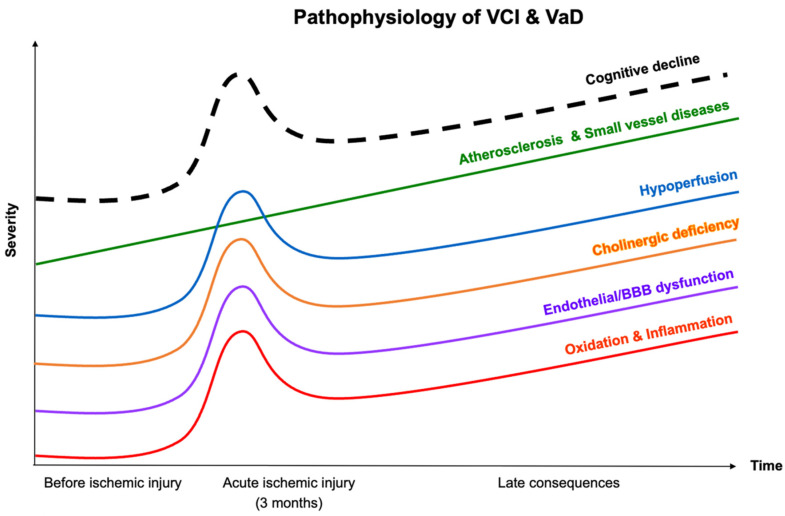
Progression of the pathophysiology of vascular cognitive impairment (VCI) or vascular dementia (VaD). BBB, blood–brain barrier.

**Figure 2 ijms-23-11067-f002:**
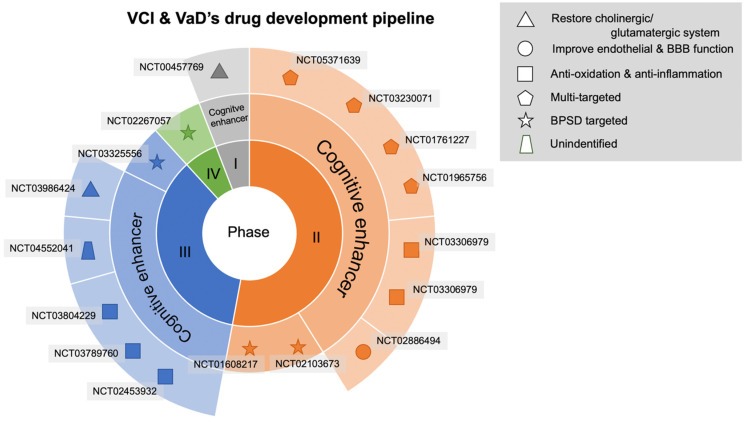
Drug development pipeline in VCI or VaD treatment. The agents are displayed using specific shapes corresponding to their pathophysiological mechanisms. BBB, blood–brain barrier; BPSD, behavioral and psychological symptoms of dementia.

**Figure 3 ijms-23-11067-f003:**
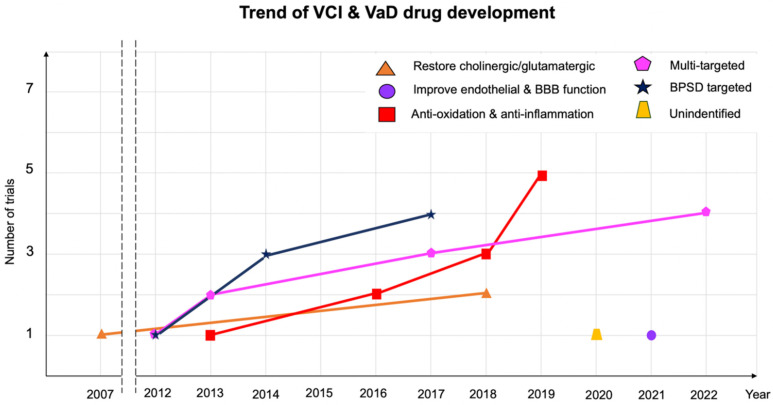
VCI and VaD drug development between 2007 and 2022 (registered on ClinicalTrials.gov accessed on 20 May 2022).

**Figure 4 ijms-23-11067-f004:**
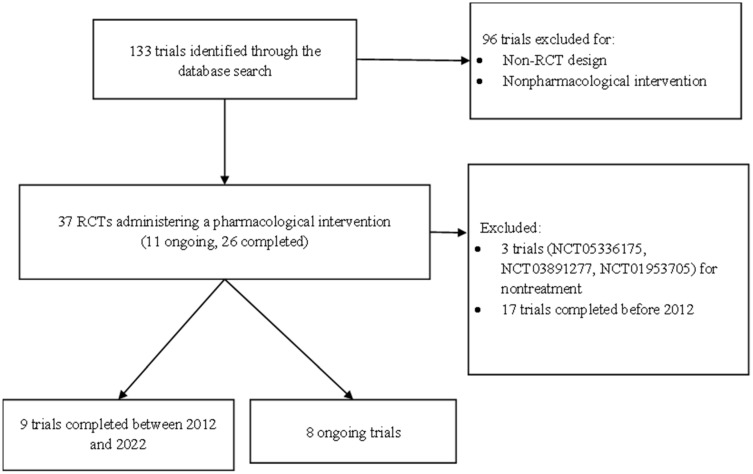
Flowchart of trial selection from a search of ClinicalTrials.gov accessed on 20 May 2022.

**Table 1 ijms-23-11067-t001:** Therapeutic agents for VCI and VaD treatment in phase III testing.

Drug	Target Type	Mechanism of Action	Therapeutic Purpose	NCT Number	Status	Country	Start Year	Estimated Year of Completion
Akatinol memantine	Enhance the effect of the glutamatergic system	**N-methyl-D-aspartate****(NMDA) glutamate receptor agonist** Neurotransmitter based Antagonizes glutamate toxicity Neuroprotection	Cognitive function enhancer	NCT03986424	Active, not recruiting	Russia	2018	2022
Butylphthalide soft capsules	Antioxidation and anti-inflammation	***DL-3-n-butylphthalide*** Improves the oxidative stress response of the nervous system Inhibits neuronal apoptosis and autophagy Regulates the function of the central cholinergic system Promotes neuroplasticity	Cognitive function enhancer	NCT03804229	Recruiting	China	2019	2024
SaiLuoTong capsules (traditional Chinese medicine)	Antioxidant and anti-inflammatory	***Ginkgo biloba***—anti-inflammation and neurogenesis ***Ginsenoside Rg1***—antioxidation and stress-induced neuronal apoptosis ***Saffron***—scavenging of oxygen free radicals	Cognitive function enhancer	NCT03789760	Recruiting	China	2019	2023
Prospecta	Unknown	Unidentified	Cognitive function enhancer	NCT04552041	Active, not recruiting	Russia	2020	2023
Tianzhi granules (traditional Chinese medicine)	Antioxidant and anti-inflammatory	Reduces oxidative stress Mitigates apoptosis and necrosis	Cognitive function enhancer	NCT02453932	Completed	China	2013	2017
Pimavanserin	Neuropsychiatric	Antipsychotic	Reduces the relapse of psychotic symptoms	NCT03325556	Completed	Multiple countries	2017	2019

**Table 2 ijms-23-11067-t002:** Therapeutic agents for VCI and VaD treatment in phase I/II testing.

Drug	Target Type	Mechanism of Action	Therapeutic Purpose	NCT Number	Status	Country	Phases	Start Year	Estimated Year of Completion
Donepezil	Restores the function of the central cholinergic system	**Cholinesterase inhibitor** Inhibits acetylcholinesterase (AChE) and increases acetylcholine release	Cognitive function enhancer	NCT00457769	Active, not recruiting	United States	Phase I	2007	2021
N-acetylcysteine	Antioxidant and anti-inflammatory	**Glutathione precursor** Exerts antioxidant and anti-inflammatory effects	Cognitive function enhancer	NCT03306979	Recruiting	Canada	Phase II	2018	2022
CY6463	Improves endothelial and BBB dysfunction	**Central nervous system–penetrant guanylyl cyclase stimulator** Compensates for NO deficiency	Cognitive function enhancer	NCT04798989	Recruiting	United States	Phase II	2021	2022
Tianmabianchunzhigan (traditional Chinese medicine)	Multitarget	***Gastrodia elata*** Reduces inflammation Mitigates apoptosis Suppresses the formation of beta-amyloid plaques	Cognitive function enhancer	NCT05371639	Not yet recruiting	Not mentioned	Phase II	2022	2025
				NCT03230071	Completed	China	Phase II	2017	2021
Fufangdanshen tablets (traditional Chinese medicine)	Multitarget	***Tanshinone***—inhibits inducible NO synthase (iNOS) and matrix metalloproteinase 2 (MMP 2), reduces free radicals and oxidation ***Salvianolic acid***—inhibits oxidation and inflammation ***Borneol***—increases BBB permeability	Cognitive function enhancer	NCT01761227	Completed	China	Phase II	2012	2015
Delta-9-tetrahydrocannabinol (delta-THC) + acetaminophen	Neuropsychiatric	Analgesic	Treatment of pain-induced behavioral disturbances	NCT01608217	Completed	Netherland	Phase II	2012	2014
Metformin	Multitarget Atherosclerosis	**Hypoglycemic agent** Antidiabetic Reduces oxidative stress and inflammation Improves endothelial function Inhibits AChE activity	Cognitive function enhancer	NCT01965756	Completed	USA	Phase II	2013	2017
D-amino acid oxidase inhibitor	Neuropsychiatric	NMDA receptor enhancer	Treatment of behavioral disturbances Cognitive function enhancer	NCT02103673	Completed	Taiwan	Phase II	2014	2017
BAC	Antioxidant and neuroinflammatory	***Glycine max******(L.) Merr*** Reduce levels of procytokine interleukin 1-β	Cognitive function enhancer	NCT02886494	Completed	USA	Phase II	2016	2018

**Table 3 ijms-23-11067-t003:** Therapeutic agent for VCI and VaD treatment in phase IV testing.

Drug	Target Type	Mechanism of Action	Therapeutic Purpose	NCT Number	Status	Country	Phases	Start Year	Estimated Year of Completion
Paracetamol	Neuropsychiatric	Analgesic	Treatment of pain-induced depression	NCT02267057	Completed	Norway	Phase IV	2014	2016

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
