# Peer review of "Clinical Trials of New Drugs for Vascular Cognitive Impairment and Vascular Dementia"

_ijms, 2022, doi:10.3390/ijms231911067_

Round 1

Reviewer 1 Report

This is a review of the clinical trials with the drugs for vascular cognitive impairment (VCI) and Vascular Dementia (VaD) published in recent years.

The authors listed the proposed mechanisms of VCI:  atherosclerosis, cerebral small-vessel diseases, cerebral hypoperfusion, oxidative stress and inflammation, endothelial dysfunction, blood–brain barrier (BBB) disruption, cholinergic degeneration.

They searched on ClinicalTrials.gov randomized clinical trials in phases I, II, III, and IV, recently finished or ongoing for new drugs for VCI.

The authors listed all the trials, according to pathophysiological mechanism of VaD targeted by the drug and also by the study phase.

They found only 17 randomized controlled trials, 9 were completed in the past 10 years, and 8 are ongoing. A smaller number of trials were targeting VCI, comparing to Alzheimer’s disease.

They concluded that is little interest of pharmaceutical companies with regard of this research domain.

Author Response

We would like to thank you for taking the time and effort to review our manuscript.

Reviewer 2 Report

This review about clinical trials of new drugs for vascular cognitive impairment (VCI) and vascular dementia (VaD) is in right time not the least as trials in dementia research not yet have been successful in identifying drugs for Alzheimer’s disease (AD). Other approaches than focusing on AD are needed. The vascular approach is one putative interesting way of finding treatments for cognitive impairment and dementia. The authors report ongoing and planned trials for VCI and VaD at various phases after 2012 from a search of ClinialTrials.gov. They conclude that the investment of pharmaceutical companies in new drugs targeting VCI is insufficient. The review is informative. However, there is room for improvement, especially with regard to the discussion.

Page 2, 1.2. Comments on the probably most common VCI subtypes, subcortical VaD is lacking. Update the paragraph.   

Page 2, 2. Reference is lacking.

Page 4, 2.5. The cholinergic system in VaD has been studied earlier than mentioned by the authors. See for instance PMID: 2568069.  Add information.

Page 7, 3.1. Ref. 38 and 40. Are they correct? Please check references throughout the manuscript.

Page 10, 3.4. (i) The text about the cholinergic system should be moved to page 4, 2.5.
(ii) There are more recent recommendations from Canada about the use of choline esterase inhibitors. See PMID: 33209971.
(iii) Put the text about memantine in a new paragraph.

Page 11, 4. The discussion is comparatively superficial and partially a repetition of what has been reported about specific drug trials. Former reviews of RCTs for VCI is not referred to especially PMID: 28476873. It should be rewritten and notifying the aspects that are discussed in 28476873 including a comparison between different search method (se below).  

Page 13. Fig. 1. (i) “17 trials completed before 2012” is at variance with 28476873. Please clarify.
(ii) In 28476873 the search was performed by means of ALOIS. What is the result if you are using ALOIS?

Author Response

Page 2, 1.2. Comments on the probably most common VCI subtypes, subcortical VaD is lacking. Update the paragraph.   

Thank you for your suggestion. We have updated the paragraph with some words from the lines 48 to 53 as below:

“…Furthermore, as VaD combines different vascular mechanisms and changes in the brain, and has different etiologies and clinical presentations, this heterogeneity in definition can affect the outcomes of clinical trials. Therefore, subcortical (ischemic) VaD is a more homogeneous population and may be an alternative for clinical drug trials, including small vessel disease, lacunar cerebral infarction, and ischemic white matter lesions.”

Page 2, 2. Reference is lacking.

Thank you for pointing it out. We have added the relevant references in the paragraph.

Page 4, 2.5. The cholinergic system in VaD has been studied earlier than mentioned by the authors. See for instance PMID: 2568069.  Add information.

Thank you for your suggestion. We have added some words about the study in 1989 of cholinergic system disturbances in VaD patients (in lines 148 to 150).

“…Markedly disturbances of the cholineacetyl transferase were found early in a 1989 study of brain tissues among dementia patients with histories of stroke.”

Page 7, 3.1. Ref. 38 and 40. Are they correct? Please check references throughout the manuscript.

Thank you for pointing it out. The current version is correct. We have rechecked all of the references in the manuscript and cited them in the correct position now.

Page 10, 3.4. (i) The text about the cholinergic system should be moved to page 4, 2.5.
(ii) There are more recent recommendations from Canada about the use of choline esterase inhibitors. See PMID: 33209971.
(iii) Put the text about memantine in a new paragraph.

Thank you for your suggestion. We have moved the text about the cholinergic system in page 10, part 3.4 to page 4, part 2.5. 23. We also changed the recommendations of fourth Canadian Consensus Conference (CCC) to the most updated (fifth) guideline of CCC. The revised text can be found in lines 340 to 342:

“…In 2020, the fifth Canadian Consensus Conference has recommended cholinesterase inhibitors may be used to treat vascular cognitive impairment in selected patients who are explained about the benefits and harms of these drugs.”

Memantine text has been put in a distingue paragraph from Cholinesterase inhibitors text, as below:

“A Cochrane Library review of two studies of approximately 750 participants concluded that memantine confers small clinical benefits for cognitive function with low-to-moderate certainty….” (lines 349 to 362)

Page 11, 4. The discussion is comparatively superficial and partially a repetition of what has been reported about specific drug trials. Former reviews of RCTs for VCI is not referred to especially PMID: 28476873. It should be rewritten and notifying the aspects that are discussed in 28476873 including a comparison between different search method (see below).  

Thank you for your comments. We have added a paragraph comparing Smith et al’s study and our review (from lines 472 to 487).

“In this review, we focused on trials registered to ClinicalTrials.gov and did not expand our search to other databases like AMIce, a German drug information system ChicCTR, a Chinese clinical trial register; CTRI, an Indian clinical trials register, or ALOIS, a database of Cochrane Collaboration Dementia and Cognitive Impairment Group might be a limitation. A review by Smith et al. in 2017 also summarized drug development for vascular dementia but by a search on ALOIS [90]. They identified 130 RCTs from 1966 to 2016 that preceded our trials’ timeline by forty years. They reported more trials than we did since they included both pharmacologic and non-pharmacologic interventions, and not only for treatment purposes but also for preventative aspects. The authors also classified the trials based on the drugs’ therapeutic effects on VCI pathophysiology, however, they defined some different drug classes including vasodilators, neurotrophic, antithrombotic, lipid-lowering, and metabolic-based mechanisms. The differences can be explained by the different concepts about the diseases’ pathophysiology. Regarding the pipeline of VCI drug development, Smith's review stated a predominance in trials targeting perfusion enhacement via vasodilators which are popular in the first twenty decades, while later, especially from 1990 to 2016, more trials focused on testing the drugs classified as neurotransmitter modulators or multiple mechanisms of action. This finding somehow is in line with our report on emerging trials on multi-targeted agents.”

Page 13. Fig. 1. (i) “17 trials completed before 2012” is at variance with 28476873. Please clarify.

In our review, we focused on the trials currently active or finished in recent ten years. That’s why we excluded trials completed before 2012. Smith’s review collected published trials from 1966 to 2016, preceding our review timeline by forty years.

(ii) In 28476873 the search was performed by means of ALOIS. What is the result if you are using ALOIS?

In this review, we did not expand our research to ALOIS, which was listed as our research limitation. We had limited access to the ALOIS database to do a search as you suggested.

Round 2

Reviewer 2 Report

The authors have updated and checked the text  in a satisfactory way.